# Proteoglycan SPOCK1 as a Poor Prognostic Marker Promotes Malignant Progression of Clear Cell Renal Cell Carcinoma via Triggering the Snail/Slug-MMP-2 Axis-Mediated Epithelial-to-Mesenchymal Transition

**DOI:** 10.3390/cells12030352

**Published:** 2023-01-17

**Authors:** Yung-Wei Lin, Yu-Ching Wen, Chi-Hao Hsiao, Feng-Ru Lai, Shun-Fa Yang, Yi-Chieh Yang, Kuo-Hao Ho, Feng-Koo Hsieh, Michael Hsiao, Wei-Jiunn Lee, Ming-Hsien Chien

**Affiliations:** 1International Master/Ph.D. Program in Medicine, College of Medicine, Taipei Medical University, Taipei 11031, Taiwan; 2Department of Urology, School of Medicine, College of Medicine and TMU Research Center of Urology and Kidney (TMU-RCUK), Taipei Medical University, Taipei 11031, Taiwan; 3Department of Urology, Wan Fang Hospital, Taipei Medical University, Taipei 11696, Taiwan; 4Graduate Institute of Clinical Medicine, College of Medicine, Taipei Medical University, Taipei 11031, Taiwan; 5Institute of Medicine, Chung Shan Medical University, Taichung 404, Taiwan; 6Department of Medical Research, Tungs’ Taichung MetroHarbor Hospital, Taichung 435403, Taiwan; 7The Genome Engineering & Stem Cell Center, School of Medicine, Washington University, St. Louis, MO 63105, USA; 8The Genomics Research Center, Academia Sinica, Taipei 11529, Taiwan; 9Department of Medical Education and Research, Wan Fang Hospital, Taipei Medical University, Taipei 11696, Taiwan; 10TMU Research Center of Cancer Translational Medicine, Taipei Medical University, Taipei 11031, Taiwan; 11Pulmonary Research Center, Wan Fang Hospital, Taipei Medical University, Taipei 11696, Taiwan; 12Traditional Herbal Medicine Research Center, Taipei Medical University Hospital, Taipei 110301, Taiwan

**Keywords:** SPOCK1, metastasis, clear cell renal cell carcinoma, snail/slug, matrix metalloproteinase-2, epithelial-to-mesenchymal transition

## Abstract

Sparc/osteonectin, cwcv, and kazal-like domains proteoglycan 1 (SPOCK1) has been reported to play an oncogenic role in certain cancer types; however, the role of SPOCK1 in the progression of clear cell renal cell carcinoma (ccRCC) remains elusive. Here, higher SPOCK1 transcript and protein levels were observed in ccRCC tissues compared to normal tissues and correlated with advanced clinical stages, larger tumor sizes, and lymph node and distal metastases. Knockdown and overexpression of SPOCK1 in ccRCC cells led to decreased and increased cell clonogenic and migratory/invasive abilities in vitro as well as lower and higher tumor growth and invasion in vivo, respectively. Mechanistically, the gene set enrichment analysis (GSEA) database was used to identify the gene set of epithelial-to-mesenchymal transition (EMT) pathways enriched in ccRCC samples with high SPOCK1 expression. Further mechanistic investigations revealed that SPOCK1 triggered the Snail/Slug–matrix metalloproteinase (MMP)-2 axis to promote EMT and cell motility. Clinical ccRCC samples revealed SPOCK1 to be an independent prognostic factor for overall survival (OS), and positive correlations of SPOCK1 with MMP-2 and mesenchymal-related gene expression levels were found. We observed that patients with SPOCK1^high^/MMP2^high^ tumors had the shortest OS times compared to others. In conclusion, our findings reveal that SPOCK1 can serve as a useful biomarker for predicting ccRCC progression and prognosis, and as a promising target for treating ccRCC.

## 1. Introduction

Renal cell carcinoma (RCC) includes clear cell renal cell carcinoma (ccRCC), which represents the most common form of RCC at >80%, and papillary (p)RCC, the second most common subtype of RCC at 10~15% [1]. Surgical excision is the standard treatment for localized ccRCC; however, 30% of ccRCC patients have metastases at the time of diagnosis [2] and about 60% patients have metastases within the initial 2~3 years after diagnosis with localized disease [3]. Metastasis is the major cause of mortality associated with ccRCC. Until now, treatment options available for this deadly stage have been very limited [4], because the molecular mechanism of metastasis has remained unclear. Therefore, identifying new mechanisms, biomarkers, and related treatment targets is of great clinical importance for managing ccRCC patients.

Tumor metastasis is a complex multifactorial process that is considered to be a vital step in cancer progression. Matrix metalloproteinases (MMPs) are key enzymes involved in the process of cancer metastasis because they were reported to degrade the main components of the basement membrane and extracellular matrix (ECM) [5], facilitate the formation of blood vessels, and enhance cell invasion. The principal MMPs involved in cancer are MMP-2, MMP-9, and, most notably, MMP-14. Kugler et al., Kallakury et al., and Abdel Wahed et al. reported strong correlations of increased MMP-2 expression with the tumor stage, vascular invasion, and poor prognosis in patients with RCC [6,7,8]. The epithelial-to-mesenchymal transition (EMT) is a process in which epithelial cells lose their polarity and barrier integrity and develop a mesenchymal phenotype, which includes the acquisition of motility [9]. The EMT is known to play a key role in the progression of ccRCC [10]. The ECM plays a critical role in the occurrence of the EMT due to changes in adhesion between cells and the matrix. Therefore, MMPs are crucial mediators of the cancer EMT as they digest the ECM [11]. As already reported, MMP-14 expression was correlated with the extent of the renal epithelial tumor EMT and invasive capacity [12]. Therefore, regulating the EMT process and expression levels of MMPs might represent a key axis for RCC prevention and therapy.

Sparc/osteonectin, cwcv, and kazal-like domain proteoglycan 1 (SPOCK1) was initially identified in human testes and originally called “testican-1”. It is a Ca^2+^-binding matricellular glycoprotein belonging to the secreted protein acidic and rich in cysteine (SPARC) family [13]. SPOCK1 shares a similar structure with other members of the SPARC family, having an N-terminus with a follistatin-like domain, and it plays a crucial role in cell proliferation, apoptosis, adhesion, and migration [14]. Recent evidence showed that SPOCK1 overexpression was observed in several cancer types, such as prostate [15], colorectal [16], gastric [17], pancreatic [18], and lung [19] cancers, and it was involved in modulating cancer cell proliferation, apoptosis, metastasis, and drug resistance. However, to the best of our knowledge, information concerning the relationship between SPOCK1 and ccRCC, including the function, molecular mechanisms, and clinical potential, is limited.

This study attempted to investigate the role of SPOCK1 in ccRCC progression and its potential mechanisms. We demonstrated significant correlations of high expression levels of SPOCK1 in tumor tissues with advanced clinical stages, larger tumor sizes, lymph node and distal metastases, and worse prognoses in patients with ccRCC. We further found that SPOCK1 contributed to the progression of ccRCC via the Snail/Slug–MMP-2 axis-mediated EMT.

## 2. Materials and Methods

### 2.1. Materials

Dimethyl sulfoxide (DMSO) was purchased from Sigma-Aldrich (St. Louis, MO, USA). The SPOCK1 recombinant protein (2327-PI) was obtained from R&D Systems (Minneapolis, MN, USA). The antibodies we used were as follows: SPOCK1 (HPA007450), β-actin (A5441) (Sigma-Aldrich); E-cadherin (#3195), N-cadherin (#13116), vimentin (#5741), Snail (#3879), Slug (#9585), MMP-3 (#14351), MMP-2 (#4022), phosphorylated (p)-Akt (#9271) (Cell Signaling Technology, Danvers, MA, USA); MMP-9 (GTX100458) (Genetex, Hsinchu, Taiwan); MMP-14 (sc-377097) (Santa Cruz Biotechnology, Santa Cruz, CA, USA); MMP-16 (bs-1856R) (Thermo Fisher Scientific, Rockford, IL, USA); GAPDH (60004-1-Ig) (Proteintech, Rosemont, IL, USA).

### 2.2. Cell Lines and Cell Culture

Human RCC cell lines (ccRCC: Caki-1 and 786-O; pRCC: ACHN and Caki-2) with wild-type (WT) or mutant Von Hippel-Lindau (VHL) were purchased from the American Type Culture Collection (Manassas, VA, USA). Caki-1 and ACHN cells were cultured in minimum essential medium (MEM), 786-O cells were cultured in RPMI-1640 medium, and Caki-2 cells were cultured in McCoy’s 5A medium. All culture media were supplemented with 10% fetal bovine serum (FBS; Gibco BRL, Gaithersburg, MD, USA), 2 mM L-glutamine, 100 U/mL penicillin, and 100 μg/mL streptomycin, and cells were maintained in a humidified incubator containing 5% CO_2_ at 37 °C.

### 2.3. Bioinformatics Analysis

Messenger (m)RNA levels of SPOCK1 in normal and ccRCC tumor tissues were obtained from the GSE15641, GSE150404, and GSE85258 datasets of the Gene Expression Omnibus (GEO) database. The Cancer Genome Atlas (TCGA) database was used to obtain SPOCK1-normalized expression data and associated clinical data, which corresponded to the ccRCC (*n* = 533) and pRCC (*n* = 285) datasets and were all downloaded from the UCSC Xena browser (https://xena.ucsc.edu/, accessed on 30 November 2021). Box plots for SPOCK1 expression values were created with respect to the clinical stage, tumor size, and lymph node and distal metastases. The prognostic significance of SPOCK1, MMP16, MMP14, and MMP2 levels and the combined effects of these genes in patients with ccRCC were determined using a Kaplan–Meier analysis. Further, the STRING website (https://string-db.org/, accessed on 30 November 2021) was utilized to explore protein–protein interaction (PPI) networks of SPOCK1-regulated proteins. Correlations of SPOCK1 with EMT-related genes and *MMPs* in ccRCC were evaluated using the cBioPortal platform (https://www.cbioportal.org/, accessed on 28 August 2022).

### 2.4. Immunohistochemical (IHC) Staining

SPOCK1 expression levels were detected by IHC staining in established human RCC tissue microarrays (TMAs) (KD2602) (US Biomax, Rockville, MD, USA). Briefly, paraffin-embedded RCC tissue sections (4 μm) were washed in xylene to remove the paraffin, rehydrated with gradient ethanol concentrations, and washed with phosphate-buffered saline (PBS) at pH 7.2. Deparaffinized sections were then boiled in a microwave in 0.1 M citric acid buffer (pH 6.0) for antigen retrieval. The TMA was then incubated overnight at 4 °C with a primary antibody for SPOCK1 (1:100; HPA007450), then the secondary antibody at 1:200 was incubated with the slides at room temperature for 1 h. The TMA was subsequently observed with a diaminobenzidine (DAB) kit (Boster, Wuhan, China) according to the manufacturer’s instructions. Nuclei were counterstained with hematoxylin.

### 2.5. Cell Extract Preparation and Western Blot Analysis

Protein lysates were prepared as described previously [20]. Appropriate quantities of protein (30~50 μg) were separated by 10~12% sodium dodecylsulfate polyacrylamide gel electrophoresis (SDS-PAGE) and transferred onto polyvinylidene difluoride (PVDF) membranes. The membranes were then probed with indicated primary antibodies overnight at 4 °C and horseradish peroxidase-conjugated secondary antibodies for 45 min at room temperature. After washing, blots were incubated with the enhanced chemiluminescence (ECL) Western blotting reagent, and antibody-bound protein bands were detected using a chemiluminescence imaging system, MultiGel-21 (TOP BIO, New Taipei City, Taiwan). β-actin and GAPDH served as the protein loading controls.

### 2.6. Transwell Migration and Invasion Assays

Migration and invasion assays were performed according to our previous study [21]. Briefly, 2 × 10^4^ RCC cells were plated in an uncoated top chamber (24-well insert; pore size, 8 μm; Corning Costar, Corning, NY, USA) for the transwell migration assay. The invasion assay used 4 × 10^4^ RCC cells plated in the top chamber, which was coated with 30 μL of 1 mg/mL Matrigel (BD Biosciences, Bedford, MA, USA). In both assays, serum-free medium was added to the top chamber, and 1 mL of complete medium was used as a chemoattractant in the lower chamber. After 24 h of incubation, migrated or invaded cells on the lower surface of the membrane were fixed with methanol and stained with 0.5% crystal violet. Numbers of cells in at least three random microscopic fields (×100 or ×200) were counted.

### 2.7. Lentiviral Production and Infection of ccRCC Cells

The SPOCK1-expressing plasmid, pLenti-GIII-CMV-SPOCK1, and a short hairpin (sh)RNA for SPOCK1, shSPOCK1, were purchased from Applied Biological Materials (Richmond, BC, Canada) and the National RNAi Core Facility at Academic Sinica (Taipei, Taiwan), respectively. The lentiviral particle preparation and ccRCC cell lines expressing SPOCK1 or SPOCK1 shRNAs were established on the basis of a previously described protocol [15]. Briefly, 293T packaging cells were transfected with 10 μg of the pLenti-GIII-CMV-SPOCK1- or shSPOCK1-expressing plasmids together with 10 μg pCMVDR8.91 (the packaging vector) and 1 μg pMD.G (the envelope vector). After 5 h of incubation, the transfection medium was replaced with fresh culture medium for a subsequent 48 h, and lentivirus-containing medium was further collected by centrifugation at 1500 rpm. Next, ccRCC cells were infected with fresh lentivirus-containing medium (supplemented with 8 μg/mL polybrene) for 24 h and subjected to different functional assays.

### 2.8. Cell Viability and Colony-Formation Assays

Caki-1 and 786-O cells were stably infected with a virus carrying either shSPOCK1, SPOCK1, or their respective controls and seeded in 96-well plates (5 × 10^3^ cells/well) containing culture medium with 10% FBS for indicated time points and then subjected to a cell viability assay (MTS assay; Promega, Madison, WI, USA) according to the manufacturer’s instructions.

For the colony-formation assay, 10^3^ SPOCK1-overexpressing or -knockdown (KD) Caki-1 and 786-O cells were seeded in six-well dishes and then cultured under standard conditions. After 7~10 days, cells were fixed with methanol and then stained with 1% crystal violet. Numbers of colonies were manually counted using free ImageJ software (version 1.53e) (National Institutes of Health, Bethesda, MD, USA).

### 2.9. Gene Set Enrichment Analysis (GSEA)

mRNA expression data of The Cancer Genome Atlas Kidney Renal Clear Cell Carcinoma (TCGA-KIRC) were downloaded from cBioPortal (https://www.cbioportal.org/, accessed on 28 August 2022). After ranking all samples by SPOCK1’s mRNA expression, the top 30 samples were defined as the high-expression group, while the bottom 30 samples were designated the low-expression group. Genes were analyzed by GSEA software with the Molecular Signatures Database (MSigDB) hallmark gene set to identify gene enrichment pathways [22]. The number of genes was set to 500 for calculating the enrichment coefficient score and normalized enrichment score (NES). A false discovery rate (FDR) of <0.05 was considered significant enrichment.

### 2.10. Transient Transfection of DNA

pLEX-Snail and pCIneo-Slug plasmids were obtained from Dr. T.C. Kuo (National Taiwan University, Taipei, Taiwan). The MMP-2 plasmid was obtained from GeneCopoeia (Rockville, MD, USA). To overexpress Snail, Slug, and MMP-2, semiconfluent cultures of ccRCC cells in 6-mm^2^ Petri dishes were transfected with 3 μg of an empty vector (EV) or expression vector using the Lipofectamine 3000 Transfection Reagent (Invitrogen, Carlsbad, CA, USA) for 6 h according to the manufacturer’s instructions. At 48 h after transfection, cells were analyzed for invasion/migration and Snail, Slug, and MMP-2 expressions.

### 2.11. Gelatin Zymography

MMP-2 activities in conditioned medium from ccRCC cells were measured using gelatin zymography assays, as previously described [23]. Briefly, an appropriate volume of collected media was electrophoresed on 8% SDS-PAGE gels containing 0.1% gelatin. After electrophoresis, gels were subjected to two 30-minute washes with 2.5% Triton X-100 prior to the development of enzyme activity bands in reaction buffer (40 mM Tris-HCl at pH 8.0, 10 mM CaCl_2_ and 0.01% NaAzide) for 12 h at 37 °C. Resulting gelatinolytic enzymes were detected as transparent bands of digested gelatin against a Coomassie blue-stained gel background.

### 2.12. MMP-2 Promoter-Driven Luciferase Assays

The MMP-2 promoter was inserted into the pGL3-basic vector (Promega, Madison, WI, USA) to generate the MMP-2 promoter/reporter plasmid. SPOCK1-depleted or control ccRCC cells were seeded at a concentration of 5 × 10^4^ cells/well in six-well cell culture plates. Cells were then cotransfected with 1 μg of the pGL3-basic or MMP-2 promoter construct and 0.5 μg of the pRL-TK Renilla control vector (Promega) using the Lipofectamine 3000 Transfection Reagent. Firefly and Renilla luciferase activities were evaluated using a dual-luciferase reporter (DLR) assay kit (Promega). Firefly luciferase activity was adjusted to Renilla luciferase activity to control for variations in cell viability and transfection efficiency.

### 2.13. Real-Time Reverse-Transcription Quantitative Polymerase Chain Reaction (RT-qPCR)

mRNA was isolated using the Trizol reagent (Invitrogen, Carlsbad, CA, USA), and complementary (c)DNA was generated using a High-Capacity cDNA Reverse Transcription kit from Applied Biosystems (Bedford, MA, USA). Then, RT-qPCR was performed using the TOOLS 2xSYBR qPCR Mix kit (BIOTOOLS Co., Ltd., Taipei, Taiwan) according to the manufacturer’s instructions. Actin was set as the internal control. Primer sequences involved in the RT-qPCR are shown as follows: MMP2 forward: GTGCTGAAGGACACACTAAAGAAGA and reverse: TTGCCATCCTTCTCAAAGTTGTAGG; actin forward: GGCGGCACCACCATGTACCCT and reverse: AGGGGCCGGACTCGTCATACT.

### 2.14. In Vivo Orthotopic ccRCC Xenograft Model

The establishment of orthotopic ccRCC xenograft mouse models was previously described [24]. All animal experiments were performed in accordance with guidelines of the Institutional Animal Care and Use Committee (IACUC) of Wang Fang Hospital-Taipei Medical University (WAN-LAC-108-020). In brief, 5 × 10^5^ ccRCC Caki-1-mock-luciferase (control group), Caki-1-SPOCK1-luciferase, and Caki-1-sh-SPOCK1-luciferase stable cell lines were directly injected into the left renal capsule of age-matched male nonobese diabetic (NOD)-SCID mice (6~8 weeks old). The day after cancer cells’ injection, noninvasive bioluminescent imaging from the Xenogen IVIS-Spectrum system (Caliper, Xenogen, CA, USA) was used to monitor tumor size weekly. After 5 weeks, mice were sacrificed and tumor-bearing tissues were excised from the mice. Finally, tumor specimens were harvested, fixed, sectioned, and stained with hematoxylin and eosin (H&E) for further histopathological analyses. The SPOCK expression status was further checked by IHC staining.

### 2.15. Statistical Analysis

Values are presented as the mean ± standard deviation (SD). Data were analyzed using Student’s *t*-test when two groups were compared. Correlations of SPOCK1 with clinicopathologic features of RCC were examined by Pearson’s Chi-squared test. Cumulative survival was analyzed by the Kaplan–Meier method. Risk factors affecting survival were assessed by a Cox proportional hazards regression model. A *p* value < 0.05 was considered a statistically significant difference.

## 3. Results

### 3.1. SPOCK1 Transcripts and Protein Were Significantly Higher in ccRCC Tumor Tissues and Were Correlated with Advanced Stages, Larger Tumor Sizes, Tumor Metastases, and Poor Patient Prognoses

To clarify the clinical relevance of SPOCK1 in patients with RCC, we first analyzed its expression levels in 23 noncancerous tissues and 69 RCC samples (including 32 cases of ccRCC) from the GSE15641 dataset of the GEO database. There was no significant difference in SPOCK1 levels between the RCC group and the normal group (Figure 1A, left panel). However, significantly higher levels of SPOCK1 transcripts were observed in ccRCC tissues compared to normal tissues (Figure 1A, right panel, *p* = 0.019). We further analyzed correlations of SPOCK1 expression with patients’ clinicopathological characteristics and the survival rate from the TCGA-KIRC database. As shown in Figure 1B, significantly higher SPOCK1 transcripts were observed in ccRCC patients with an advanced stage (stages 3 and 4), larger tumor (T3 and T4), and lymph node and distal metastases (N1 and M1). Similar results of correlations of SPOCK1 levels with clinical stage (GSE150404) and metastasis (GSE85258) were also observed in ccRCC patients from the GEO database (Appendix A). In contrast to ccRCC, SPOCK1 expression was not correlated with any clinicopathological characteristics described above in patients with papillary (p)RCC) from TCGA (Appendix A). The resulting Kaplan–Meier plot showed that ccRCC patients with high SPOCK1 expression had poor overall survival (OS) (*p* = 0.002), disease-specific survival (DSS) (*p* < 0.001), and disease-free survival (DFS) (*p* = 0.019) (Figure 1C, left panel). In contrast, the prognostic significance of SPOCK1 was not observed in patients with pRCC (Figure 1C, right panel). Moreover, we utilized a univariate Cox regression analysis to examine the prognostic significance of clinicopathologic variables in a ccRCC cohort. We observed that SPOCK1 expression (hazard ratio (HR), 1.630; *p* = 0.002), age (HR, 1.771; *p* < 0.001), clinical stage (HR, 3.833; *p* < 0.001), tumor size (HR, 3.151; *p* < 0.001), and distal metastasis (HR, 2.143; *p* < 0.001) were all shown to have adverse impacts on OS (Table 1). Furthermore, DFS was also adversely affected by high SPOCK1 expression (HR, 3.610; *p* = 0.028), tumor size (HR, 3.401; *p* = 0.019), and an advanced stage (HR, 3.440; *p* = 0.018) (Table 2). As the contribution of SPOCK1 to survival might not be independent, we further analyzed relationships of OS and DFS with SPOCK1 expression by a multivariate analysis. The results revealed that the SPOCK1 expression level was an independent prognostic factor for OS but not for DFS in ccRCC patients (Table 1 and Table 2). In addition to analyzing mRNA levels of SPOCK1 from datasets available online, we further verified protein levels of SPOCK1 by IHC staining of a tissue microarray composed of ccRCC samples at different clinical stages. Our results showed that SPOCK1 expression was enriched in ccRCC tissues, but was very low in normal kidney and benign tumor tissues (Figure 1D). Furthermore, SPOCK1 seemed to be expressed at higher levels in advanced stages compared to early stages of ccRCC (Figure 1D, right panel). Taken together, these clinical data suggest that SPOCK1 is an independent prognostic factor affecting the survival of ccRCC patients and may play a critical role in the progression of ccRCC.

### 3.2. SPOCK1 Expression Modulates the Proliferation, Clonogenicity, and Migratory and Invasive Abilities of ccRCC Cells

To further determine the functional role of SPOCK1 in RCC, we investigated the correlation between SPOCK1 expression and the cell migratory ability. We first evaluated endogenous levels of SPOCK1 in a set of RCC cell lines including ccRCC (Caki-1 and 786-O) and pRCC (ACHN and Caki-2) cells by Western blotting and found that ccRCC cells expressed higher SPOCK1 levels compared to the pRCC cell lines (Figure 2A). Next, the migratory abilities of these cell lines were further checked. We found that these cell lines harbored different migratory abilities, and Caki-1 and 786-O cells exhibited relatively high migratory abilities (Figure 2B). We next evaluated the function of SPOCK1 in modulating cell migration and invasion, two fundamental steps of tumor metastasis, in these ccRCC cell lines. SPOCK1-KD was performed by three specific short hairpin (sh)RNAs to establish stable KD of endogenous SPOCK1 in Caki-1 and 786-O cells (Figure 2C). We found that SPOCK1-KD significantly attenuated the migratory ability of both ccRCC cell lines (Figure 2D). In comparison, we further overexpressed SPOCK1 in Caki-1 and 786-O cells (Figure 2E) and evaluated their cell-invasive abilities. Indeed, SPOCK1 overexpression significantly promoted the invasive abilities of both ccRCC cell lines (Figure 2F). Next, we determined the impact of changing SPOCK1 expression on the proliferation rates of Caki-1 and 786-O cells. We found that KD (lower panel) and overexpression (upper panel) of SPOCK1 suppressed and promoted the proliferation of both ccRCC cell lines, respectively (Figure 2G). We further examined the effect of SPOCK1 on long-term growth (7~10 days) of Caki-1 and 786-O cells using a colony formation assay. As shown in Figure 2H, numbers of colonies of both ccRCC cell lines increased and decreased following overexpression and KD of SPOCK1, respectively. These results indicated that SPOCK1 may be deeply involved in ccRCC progression through inducing the growth and motility of cells.

### 3.3. GSEA Reveals the EMT to Be Positively Associated with SPOCK1

To decipher the mechanism underlying SPOCK1-induced ccRCC progression, a GSEA based on the TCGA-KIRC dataset was performed. In the SPOCK1-high group, Hallmark hypoxia, the EMT, and interleukin (IL)-6/Janus kinase (JAK)/signal transduction and activator of transcription 3 (STAT3) signaling were the top three positively enriched pathways (NES > 2.0, *p* < 0.05) (Figure 3A). The EMT is recognized as a process where epithelial cells take on characteristics of mesenchymal cells and play key roles in tumor metastasis and development. As shown in Figure 3B, SPOCK1 expression levels were positively and strongly related with EMT-associated gene signatures in TCGA-KIRC (NES = 2.1, *p* < 0.001). Moreover, human ccRCC samples retrieved from TCGA using the cBioPortal platform showed that SPOCK1 gene expression was positively correlated with mesenchymal-related genes, CDH2 (N-cadherin), FN1 (fibronectin), and VIM (vimentin), and negatively correlated with epithelial-related genes, CDH1 (E-cadherin), OCLN (occludin), and TJP1 (tight junction protein 1) (Figure 3C), suggesting that SPOCK1 might modulate the progression of ccRCC via EMT induction. Indeed, SPOCK1 depletion by shRNAs in Caki-1 and 786-O cells led to EMT suppression, as evidenced by upregulation of the epithelial marker, E-cadherin, and downregulation of the mesenchymal markers, N-cadherin and vimentin, and the EMT-promoting transcription factors, Snail and Slug (Figure 3D and Appendix A). In contrast, expressions of epithelial and mesenchymal markers decreased and increased, respectively, after SPOCK1 overexpression in Caki-1 cells (Figure 3E). Next, to determine whether Snail family members play critical roles in SPOCK1-modulated cell motility, we overexpressed Snail or Slug in Caki-1 cells to reverse SPOCK1-KD-mediated downregulation of the Snail family (Figure 3F), and inhibition of cell invasion imposed by SPOCK1 depletion was also significantly reversed (Figure 3G). Taken together, these data indicate that SPOCK1 modulates ccRCC progression via inducing the Snail family-mediated EMT. Previous studies have mentioned that the SPOCK1-mediated EMT and tumor progression via activation of the PI3K/Akt signaling pathway in various cancer types [25,26]. Herein, we also observed that treatment of ccRCC cells with a SPOCK1 recombinant protein (rSPOCK1) resulted in activation of Akt (Appendix A).

### 3.4. SPOCK1 Regulates ECM Remodeling via MMP-14/MMP-16-Mediated MMP-2 Activation and Expression

In addition to the EMT, SPOCK1 was reported to be an ECM proteoglycan, which directly or indirectly regulates ECM remodeling, thus affecting tumor progression [27]. Data from a STRING database analysis revealed PPI networks for SPOCK1 (Figure 4A, left panel). We found that MMP-14 and MMP-16 were two of the top 10 interactors (Figure 4A, right panel). Both MMPs which belong to membrane-type (MT) MMPs were reported to be involved in breaking down the ECM via inducing activation or expression of MMP-2 or MMP-9 [28,29,30]. We next screened the effects of SPOCK1 on these proteases and found that expressions of MMP-14, MMP-16, and its downstream targets, MMP-2 and MMP-9, were considerably downregulated after SPOCK1-KD in Caki-1 cells (Figure 4B). In contrast, overexpression of SPOCK1 induced upregulation of MMP-2 and MMP-9 in the same cells (Appendix A). In contrast to Caki-1 cells, Western blot data showed that only MMP-14, MMP-16, and MMP-2 were consistently suppressed by SPOCK1-KD in 786-O cells (Figure 4B). Moreover, treatment of Caki-1 cells with the rSPOCK1 for 24 or 48 h also resulted in upregulation of MMP-14/MMP-16 and MMP-2 (Figure 4C) and increased cell motility (Appendix A) in Caki-1 cells. Induction of MMP-2 by rSPOCK1 was also observed in 786-O cells (Appendix A). In addition to protein expression, MMP-2 proteolytic activity was upregulated and downregulated in SPOCK1-overexpressing and SPOCK1-depleted ccRCC cells, respectively, compared to parental cells (Figure 4D and Appendix A). To test the importance of MMP-2 expression in SPOCK1-promoted cell invasion, we rescued MMP-2 in SPOCK1-KD Caki-1 cells by transfection of an MMP-2-expressing plasmid and found that MMP-2 overexpression significantly rescued invasion suppression imposed by SPOCK1-KD in Caki-1 cells (Figure 4E). Taken together, these results suggest that SPOCK1 may regulate ECM remodeling to promote ccRCC invasion via the MMP-14/MMP-16-MMP-2 axis. From the same aforementioned TCGA dataset, we found that SPOCK1 expression in ccRCC was significantly correlated with expressions of MMP14, MMP16, and MMP2 (Figure 4F, upper panel). MMP2 expression was also significantly correlated with MMP14 and MMP16 expressions in ccRCC (Figure 4F, lower panel). Moreover, ccRCC patients with SPOCK1^high^/MMP14^high^, SPOCK1^high^/MMP16^high^, or SPOCK1^high^/MMP2^high^ all had the shortest survival times compared to other groups including SPOCK1^low^/MMP^low^, SPOCK1^high^/MMP^low^, and SPOCK1^low^/MMP^high^ (Figure 4G). These clinical data imply that the SPOCK1-regulated MMP-14/MMP-16-MMP-2 axis is associated with a poor prognosis in patients with ccRCC.

### 3.5. Crosstalk between MMP-2 and the Snail Family Is Involved in the Process of SPOCK1-Mediated EMT

Recent studies indicated that Snail can mediate the upregulation of several gene expressions of MMPs in cancers such as MMP-2, MMP-9, and MMP-14, and these proteases were reported to be promoters and mediators of EMT processes in cancers [30,31,32,33]. Herein, we observed that KD and overexpression of SPOCK1 in Caki-1 cells inhibited and promoted MMP2 mRNA expression, respectively (Figure 5A). Downregulation of MMP2 promoter activity was also observed in SPOCK1-depleted Caki-1 cells (Figure 5B). To test the importance of Snail family expression in SPOCK1-mediated MMP-2 expression, we rescued Snail or Slug in SPOCK1-KD Caki-1 cells by transfection of a Snail- or Slug-expressing plasmid. We observed that Snail or Slug overexpression significantly rescued the MMP-2 mRNA and protein suppression imposed by SPOCK1-KD (Figure 5C,D). Taken together, these results suggest the dependence of the Snail family on SPOCK1-regulated MMP-2 transactivation in Caki-1 cells. Indeed, we also found that MMP2 (MMP-2) expression exhibited high correlations with SNAI1 (Snail) and SNAI2 (Slug) expressions in ccRCC tissues (Figure 5E). Moreover, SPOCK1-KD-mediated downregulation of N-cadherin and vimentin and upregulation of E-cadherin were dramatically reversed by overexpressing MMP-2 (Figure 5F), indicating that Snail family-mediated MMP-2 expression participates in SPOCK1-induced EMT progression. In addition to MMP-2, we surprisingly observed that the downregulation of MMP-16 and SPOCK1 itself imposed by SPOCK1-KD was also reversed by overexpressing Snail or Slug in ccRCC cells (Appendix A).

### 3.6. SPOCK1 Promotes Tumorigenicity and the Invasive Ability of ccRCC Cells in an Orthotopic Mouse Model

We next examined the in vivo effects of SPOCK1 expression on ccRCC tumor progression. Herein, we established a ccRCC-bearing model by orthotopically transplanting luciferase-tagged cells, Caki-1-mock-luciferase, Caki-1-SPOCK1-luciferase, or Caki-1-shSPOCK1-luciferase, into NOD-SCID mice. The effects of SPOCK1 expression on tumor growth were further monitored weekly by bioluminescence imaging (Figure 6A). From in vivo photon emission detection, we found that SPOCK1 overexpression promoted and SPOCK1-KD attenuated tumor growth compared to the control group (Figure 6B). In addition to tumorigenicity, we further analyzed the invasive ability of tumor cells through observing the interface between orthotopic tumors and the renal parenchyma in tumor-implanted kidney. We observed that tumors derived from SPOCK1-KD group showed an expansive pattern after 5-week implantation. By contrast, tumors derived from SPOCK1-overexpressed group showed a more infiltrative invasion pattern (Figure 6C). Consistent with our in vitro findings, IHC staining of SPOCK1 and vimentin in tumor tissues from Caki-1-SPOCK1- and Caki-1-shSPOCK1-injected mice showed the upregulation and downregulation of both proteins, respectively, compared to control mice (Figure 6D).

## 4. Discussion

Metastasis is still a crucial impediment to the effective treatment of ccRCC patients, despite significant progress having been made in surgical care, chemotherapy, and targeted therapy over the past few decades. Therefore, identifying novel target molecules and the underlying mechanisms involved in cell motility and invasion is urgently needed to understand ccRCC metastasis, with the purpose of identifying novel therapeutic approaches. SPOCK1 is a proteoglycan protein and was reported to play vital roles in cell proliferation and motility in various cancer types, such as pancreatic cancer [18], gliomas [34], prostate cancer [15], gastric cancer [17], and so on. Recently, Su et al. first indicated that SPOCK1 was overexpressed in RCC tissues by single-cell RNA sequencing [35], and suggested that SPOCK1 could become a clinically useful candidate if more attention is paid to its diagnostic, prognostic, and therapeutic value. Herein, our clinical association study found that the SPOCK1 gene and protein were highly expressed in ccRCC tissues compared to normal kidney tissues. Moreover, upregulation of SPOCK1 was significantly associated with advanced clinical stages, larger tumor sizes, lymph node and distal metastases, and shorter OS, DSS, and DFS times of ccRCC patients. Furthermore, a Cox proportional hazard regression analysis identified SPOCK1 as an independent factor for predicting a poor prognosis. Taken together, the clinical analyses of SPOCK1 indicate that SPOCK1 may play a critical role in ccRCC progression and could serve as a biomarker for early detection and precise prognoses.

In this study, a series of in vitro and in vivo assays showed that cancer cell proliferative and metastatic capabilities were significantly inhibited by KD and enhanced by the overexpression of SPOCK1, suggesting its role in tumor growth and motility of ccRCC. After being identified, previous studies reported that SPOCK1 functions as a metastasis-related gene in several cancers, which promotes the EMT of cancer cells, resulting in distant cancer metastasis [36]. Our bioinformatics analysis also indicated that EMT-related mechanisms underlie the functional role of SPOCK1 in ccRCC. Moreover, we found that SPOCK1 expression was significantly correlated with expressions of mesenchymal-related markers and inversely correlated with expressions of epithelial-related markers in 512 ccRCC samples retrieved from TCGA, implying that SPOCK1 could promote ccRCC invasion by regulating the EMT process. Actually, our results showed that SPOCK1-KD suppressed the EMT progression in ccRCC cells, as evidenced by an increase in the epithelial marker, E-cadherin, and decreases in the mesenchymal markers, N-cadherin and vimentin, as well as the EMT-promoting transcription factors of Snail and Slug. Conversely, SPOCK1 overexpression yielded opposite effects on ccRCC cells. Moreover, overexpressing Snail or Slug in Caki-1 cells significantly reversed the inhibitory effect of SPOCK1-KD on expressions of Snail family members and the invasive ability, suggesting that SPOCK1 can drive the Snail family-mediated EMT, resulting in ccRCC metastasis.

PI3K/Akt was reported to play a crucial role in regulating expressions of Snail family members through multiple mechanisms in cancers. For instance, Julien et al. indicated that activation of nuclear factor (NF)-κB by Akt upregulates Snail expression in SCC-15 oral cancer cells [37]. On the other hand, activation of PI3K/Akt can phosphorylate glycogen synthase kinase (GSK)-3β to promote GSK-3β ubiquitination and degradation and further maintain the stability of Snail and Slug in prostate [38] and colorectal cancers [39]. Previous studies indicated that the SPOCK1-mediated EMT regulates proliferation and invasion via activation of the PI3K/Akt signaling pathway in several cancer types such as colorectal [26], gallbladder [25], pancreatic [18], and brain [34] cancers. Our present study also demonstrated that rSPOCK1 treatment could induce Akt activation in Caki-1 and 786-O cells, suggesting that PI3K/Akt signaling activation by SPOCK1 may be one of the causes for SPOCK1-mediated upregulation of Snail family members which induces ccRCC cell motility. However, the effects of SPOCK1 on GSK-3β degradation and NF-κB activation in ccRCC cells need to be further investigated in the future. In addition to Akt-regulated expressions of Snail family members in cancers, Snail and Slug were also reported to regulate Akt activation through transcriptional inhibition of the Akt suppressor, phosphatase and tensin homologue (PTEN) [40,41]. Whether SPOCK1-induced Snail family members can provide feedback which induces Akt activity by targeting PTEN should be further evaluated in the future.

ECM and matrix-degrading proteases participate in regulating tumor cell functions, including proliferation, survival, migration, and invasion. During the EMT, tumor cells degrade the basement membrane and gain access to the matrix by extending pseudopodia. Thus, matrix-degrading proteases, such as MMP-2, MMP-9, MMP-14 (MT1-MMP), and MMP-16 (MT3-MMP), were reported to be required for EMT progression of different cancer types [42,43,44]. SPOCK1 was reported to be a regulator of the ECM and is crucial for maintaining the process of tumor ECM dynamic homeostasis through regulating the activities and expressions of MMPs such as MMP-2, MMP-3, and MMP-9 in glioma, liver cancer, and prostate cancer [27,34,45,46]. Compared to other tumor types, little information is available on the effects of SPOCK1 on matrix-degrading proteases in ccRCC. Our present results showed downregulation of MMP-2, -3, and -9 in SPOCK1-depleted Caki-1 cells, but only MMP-2 downregulation was observed in SPOCK1-depleted 786-O cells, suggesting that targeting of MMP-2 by SPOCK1 might be a general phenomenon in ccRCC cells. In addition, enzyme activity of MMP-2 and expressions of its upstream activators, MMP-14 and MMP-16, increased in SPOCK1-overexpressing and decreased in SPOCK1-KD ccRCC cells. Moreover, the outside-in effects of secreted SPOCK1 on expressions of MMP-14/MMP-16 and MMP-2 were demonstrated by treating ccRCC cells with rSPOCK1. We further found that MMP-2 overexpression rescued the EMT and invasion suppression imposed by SPOCK1-KD in Caki-1 cells. Taken together, these results indicate that secreted SPOCK1 promotes the EMT to increase the invasive ability of ccRCC via inducing the MMP-14/MMP-16-MMP-2 axis. In patients with ccRCC, positive correlations of SPOCK1 with MMP-14, MMP-16, and MMP-2 were also observed in tumor tissues, and a combination of high SPOCK1 and high MMP2 expressions revealed the worst prognosis compared to others, suggesting that the SPOCK1-MMP-2 axis may thus be a specific biomarker for forecasting EMT-regulated invasion of ccRCC cells and be a valuable therapeutic target for treating ccRCC patients.

Snail and Slug are DNA-binding proteins which can shuttle between the cytoplasm and nuclei to bind an E-box located on the promoter upstream of E-cadherin to suppress its expression, thereby triggering the EMT [47]. In addition to the EMT regulated by Snail, nuclear Snail was also reported to play an another key role in regulating cell invasion by inducing increases in MMPs, such as MMP-1, MMP-2, MMP-7, MMP-9, and MMP-14, in ovarian, oral, and liver cancers [32,48,49]. Our present study showed that SPOCK1-KD suppressed mRNA and promoter activities of MMP-2 in Caki-1 cells, and overexpression of Snail or Slug significantly reversed the SPOCK1-KD-mediated inhibition of MMP-2 mRNA and protein levels, indicating that SPOCK1 might transactivate MMP-2 through upregulating Snail family members. Moreover, MMP-16 downregulation induced by SPOCK1-KD was also reversed by overexpressing Snail or Slug, indicating that SPOCK1-triggered MMP-14 or MMP-16 expression might also occur through upregulating the Snail family. To our surprise, overexpression of Snail family members also reversed SPOCK1 expression in SPOCK1-KD Caki-1 and 786-O cells, suggesting that Snail/Slug expression induced by SPOCK1 might exhibit positive feedback regulation on SPOCK1, and this issue should be further confirmed in the future.

## 5. Conclusions

In summary, our study revealed for the first time that the SPOCK1 expression is associated with advanced stages, larger tumor sizes, and lymph node/distal metastases, and it was identified as an independent predictive marker for poor prognoses in patients with ccRCC. Both in vitro and in vivo assays showed that SPOCK1 had strong tumorigenic functions. We further propose that SPOCK1 can induce the Snail family-mediated EMT via the MMP-14/MMP-16-MMP-2 axis in ccRCC cells, which results in metastasis; the mechanism is schematically illustrated in Figure 7. Our results indicate that SPOCK1 represents an exciting area for potential therapeutic and preventive strategies for patients with ccRCC.

## Figures and Tables

**Figure 1 cells-12-00352-f001:**
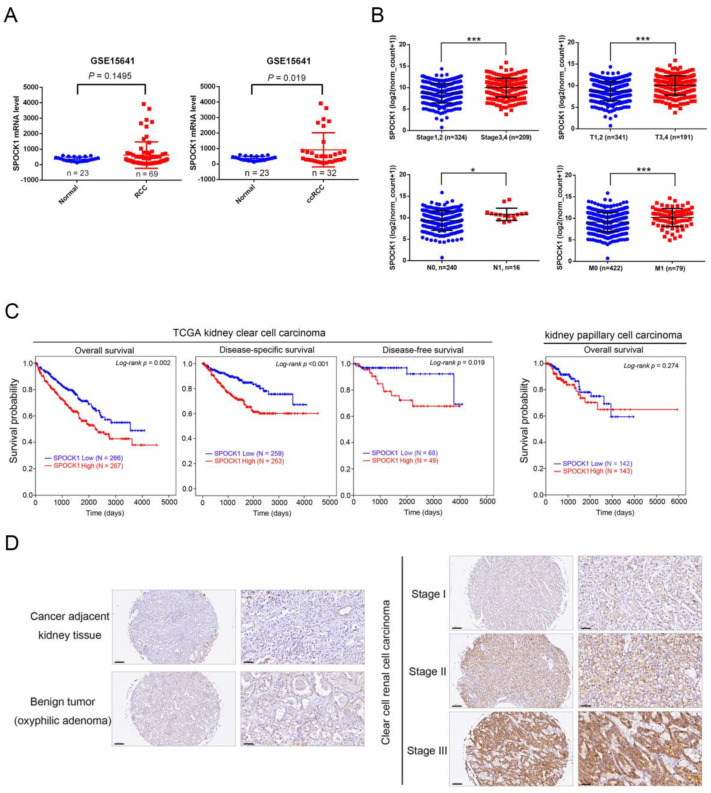
Clinical significance of SPOCK1 in clear cell renal cell carcinoma (RCC; ccRCC). (**A**) SPOCK1 gene expression levels in RCC specimens (*n* = 69), ccRCC specimens (*n* = 32), and normal tissue samples (*n* = 23) were measured by Affymetrix oligonucleotide arrays obtained from the GEO (GSE15641). (**B**) SPOCK1 gene expression levels in ccRCC samples from TCGA were compared according to the clinical stage, tumor size (T stage), lymph node metastasis (N stage), and distal metastasis (M stage). Statistical significance was analyzed by a *t*-test. * *p* < 0.05, *** *p* < 0.001. (**C**) Kaplan–Meier curves for survival of patients with ccRCC or papillary (p)RCC, as categorized according to high or low expression of SPOCK1. The *p* value indicates a comparison between patients with SPOCK1^high^ and SPOCK1^low^. The ccRCC and pRCC datasets were retrieved from TCGA. (**D**) SPOCK1 protein expression levels in ccRCC specimens and adjacent normal tissue samples or benign tumor (oxyphilic adenoma) samples were measured by IHC staining. The right panels are the enlarged images of left panels. Scale bars of left panel and right panel are 200 and 100 µM, respectively.

**Figure 2 cells-12-00352-f002:**
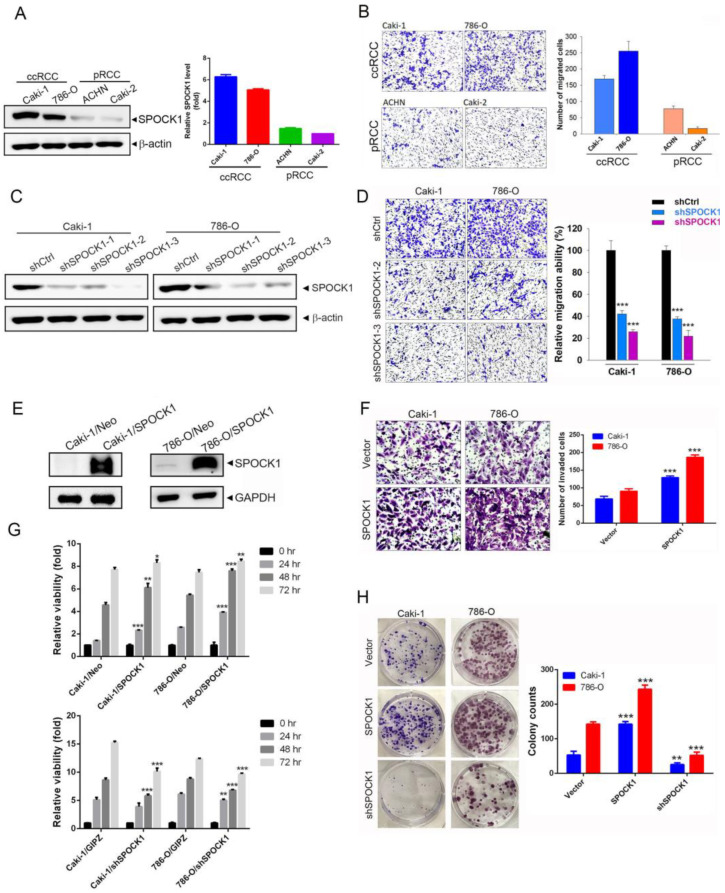
SPOCK1 overexpression promotes the proliferation, clonogenicity, migration, and invasion of clear cell renal cell carcinoma (ccRCC) cells. (**A**) Endogenous SPOCK1 protein levels were detected using a Western blot analysis of ccRCC (Caki-1 and 786-O) and papillary (p)RCC (ACHN and Caki-2) cells. (**B**) The migratory ability of ccRCC and pRCC cells was examined by a transwell migration assay. (**C**) Knockdown efficiencies of three SPOCK1 shRNAs were determined by Western blotting in Caki-1 and 786-O cells. (**D**) Migratory abilities of SPOCK1-KD Caki-1 and 786-O cells were evaluated by a transwell migration assay. (**E**) SPOCK1 was overexpressed in Caki-1 and 786-O cells as determined by Western blotting. (**F**) Invasive abilities of SPOCK1-overexpressing Caki-1 and 786-O cells were determined by a Matrigel invasion assay. (**D**,**F**) Left panel: representative photomicrographs. Right panel: Data are presented as the mean ± SD of three independent experiments. *** *p* < 0.001, compared to control cells. (**G**,**H**) Proliferation rates and colony-forming abilities of SPOCK1-manipulated Caki-1 and 786-O cells were measured by performing MTS (**G**) and colony-formation (**H**) assays, respectively. Left panel of (**H**): representative photomicrographs. Data from (**G**) and (**H**) are presented as the mean ± SD of three independent experiments. * *p* < 0.05, ** *p* < 0.01, *** *p* < 0.001, compared to control cells.

**Figure 3 cells-12-00352-f003:**
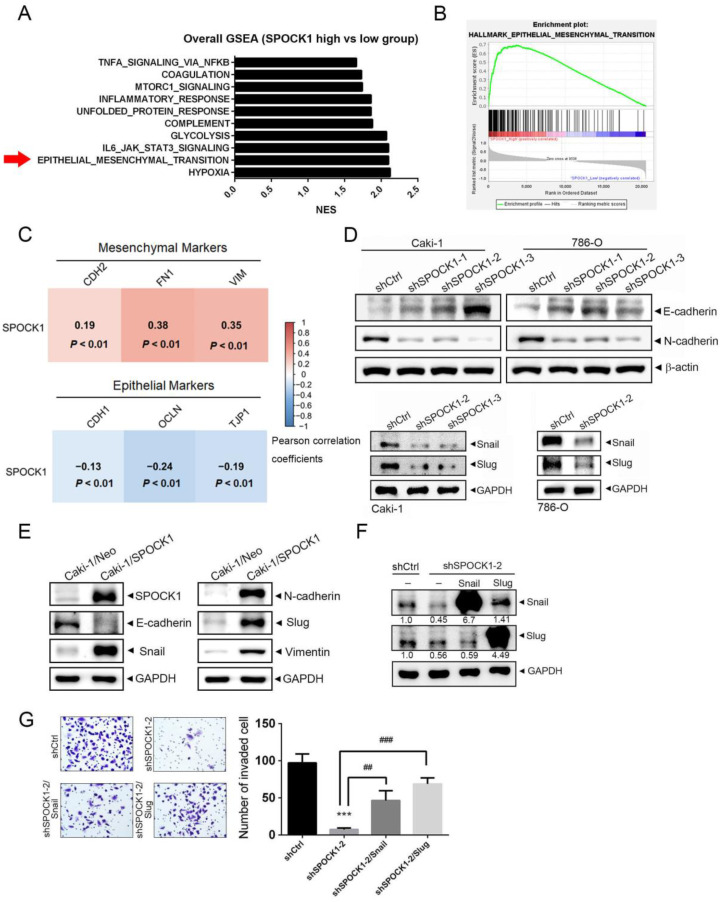
SPOCK1 promotes the epithelial-to-mesenchymal transition (EMT) in clear cell renal cell carcinoma (ccRCC) cells. (**A**) The gene set enrichment analysis (GSEA) was carried out using Hallmark gene sets from the Molecular Signature Database (http://software.broadinstitute.org/gsea/msigdb/index.jsp, accessed on 28 August 2022). Statistically significant signatures were selected (with a false detection rate (FDR) of <0.05) and placed in order of the normalized enrichment score (NES), which represents the strength of the relationship between the phenotype and gene signature. Black bars indicate pathways enriched in the high-SPOCK1-expression group. (**B**) GSEA of TCGA KIRC dataset showed an enrichment of gene signatures associated with the EMT in the high-SPOCK1-expression group. (**C**) The correlations between the gene expression of SPOCK1 and EMT markers are demonstrated in the correlation plot. The RNA sequencing data of TCGA KIRC patients were utilized to perform this analysis. Pearson correlation was conducted to evaluate the relation between SPOCK1 and EMT markers. The correlation coefficients and *p* values between EMT markers and SPOCK1 are indicated in each square. The scale bar represents the degree of correlation values. (**D**,**E**) Caki-1 or 786-O cells were infected with a lentivirus carrying either SPOCK1 shRNAs (**D**), a SPOCK1-expressing vector (**E**), or the respective control vector and subjected to a Western blot analysis to determine expressions of EMT-related regulators (E-cadherin, N-cadherin, vimentin, Snail, and Slug). Quantitative results of the indicated proteins were adjusted to GAPDH or *β*-actin protein levels. (**F**,**G**) A Snail-expressing plasmid (pLEX-Snail) or Slug-expressing plasmid (pCIneo-Slug) was transfected into Caki-1 cells expressing shCtrl or shSPOCK1 as indicated and subjected to a Western blot analysis (**F**) and Matrigel invasion assay (**G**). Quantitative results of Snail and Slug were adjusted to the GAPDH protein level. Values are presented as the mean ± SD of three independent experiments. *** *p* < 0.001, compared to the control group. ^##^
*p* < 0.01 and ^###^
*p* < 0.001 compared to the SPOCK1-KD-only group.

**Figure 4 cells-12-00352-f004:**
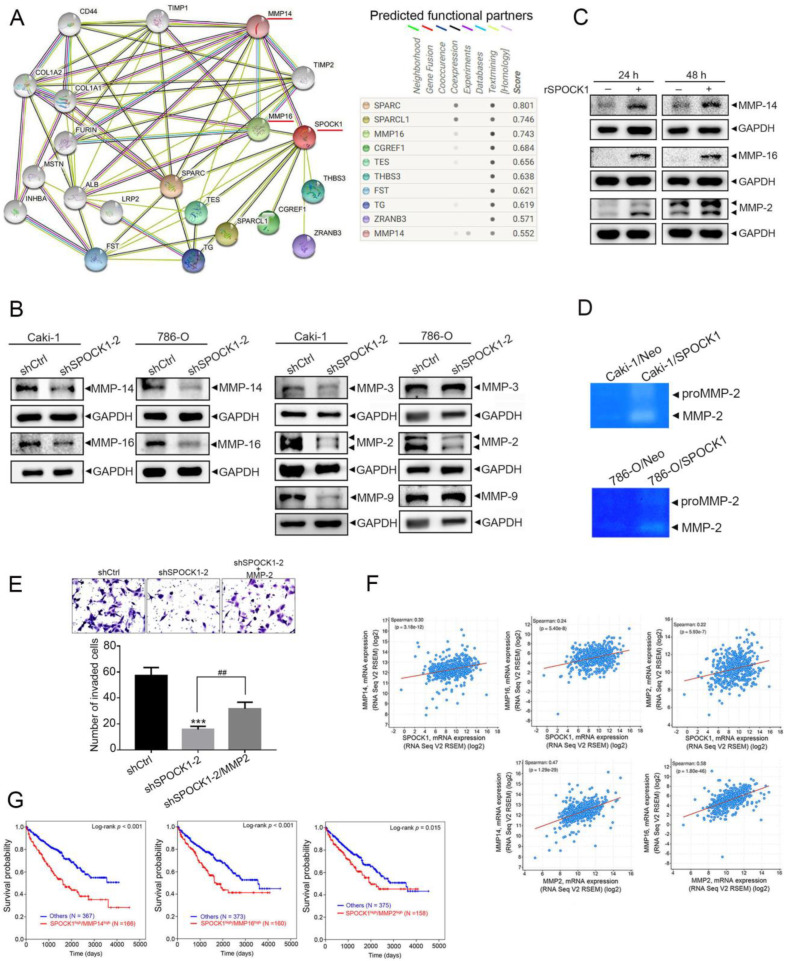
SPOCK1 regulates motility via inducing matrix metalloproteinase (MMP)-14/MMP-16-mediated MMP-2 activation in clear cell renal cell carcinoma (ccRCC) cells. (**A**) SPOCK1 protein–protein interaction network of 10 differentially expressed genes from the STRING database. (**B**) Caki-1 and 786-O cells were infected with a lentivirus carrying either SPOCK1 shRNA or shCtrl and subjected to a Western blot analysis to determine expressions of MMP-16, MMP-14, MMP-9, MMP-3, and MMP-2. (**C**) Treatment of Caki-1 cells with or without the recombinant SPOCK1 protein (rSPOCK1, 1 ng/mL) for 24 and 48 h, and expressions of MMP-16, MMP-14, and MMP-2 were detected by Western blotting. Quantitative results of indicated MMP proteins were adjusted to GAPDH protein levels. (**D**) MMP-2 activity of SPOCK1-overexpressing Caki-1 and 786-O cells was detected by a gelatin zymographic assay. (**E**) An MMP-2-expressing plasmid was transfected into Caki-1 cells expressing shCtrl or shSPOCK1 as indicated and subjected to transwell invasion assays. Top: Representative photomicrographs. Bottom: Quantitative results of the invasion assay are shown. *** *p* < 0.001, compared to the control group. ^##^
*p* < 0.01 compared to the SPOCK1-KD only group. (**F**) Correlation analysis of the KIRC database (TCGA, PanCancer Atlas) using cBioPortal revealed positive correlations of SPOCK1 or MMP2 with mRNA levels of the indicated MMP genes. (**G**) Combined expressions of high SPOCK1 and high MMP14, MMP16, or MMP2 were correlated with the overall survival of patients with ccRCC. The *p* value refers to a comparison between SPOCK1^high^/MMP^high^ and other groups (SPOCK^high^/MMP^low^, SPOCK^low^/MMP^high^, or SPOCK^low^/MMP^low^).

**Figure 5 cells-12-00352-f005:**
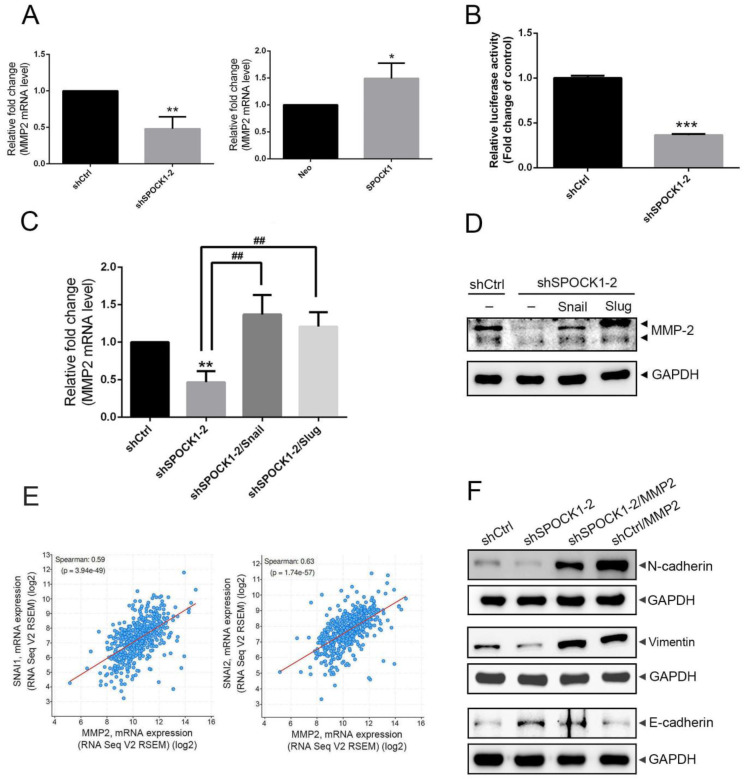
SPOCK1-induced Snail family-mediated transactivation of matrix metalloproteinase (MMP)-2 in clear cell renal cell carcinoma (ccRCC) cells. (**A**,**B**) Caki-1 cells were infected with a lentivirus carrying either shSPOCK1, SPOCK1, or their respective controls and subjected to a real-time qPCR analysis to analyze mRNA expression of MMP2 (**A**) and an MMP-2 promoter reporter assay to analyze the promoter activity of MMP-2 (**B**). Quantitative results of MMP2 mRNA levels were adjusted to actin mRNA levels. Values are presented as the mean ± SD of three independent experiments. * *p* < 0.05, ** *p* < 0.01, *** *p* < 0.001, compared to the control group. (**C**,**D**) Snail, Slug, shSPOCK1, and their respective control vectors were overexpressed in Caki-1 cells as indicated. Expression levels of MMP2 mRNA and MMP-2 proteins were detected by real-time qPCR (**C**) and Western blot analyses (**D**), respectively. Quantitative MMP-2 levels were adjusted to GAPDH (protein) or actin (mRNA) levels. ** *p* < 0.01 compared to the control group. ^##^
*p* < 0.01 compared to the SPOCK1-KD only group. (**E**) Correlation analysis of the KIRC database (TCGA, PanCancer Atlas) using cBioPortal revealed positive correlations between MMP2 and mRNA levels of SNAl1 or SNAl2 genes. (**F**) An MMP-2-expressing plasmid was transfected into Caki-1 cells expressing shCtrl or shSPOCK1 as indicated, and expression levels of EMT-related proteins were detected by a Western blot analysis.

**Figure 6 cells-12-00352-f006:**
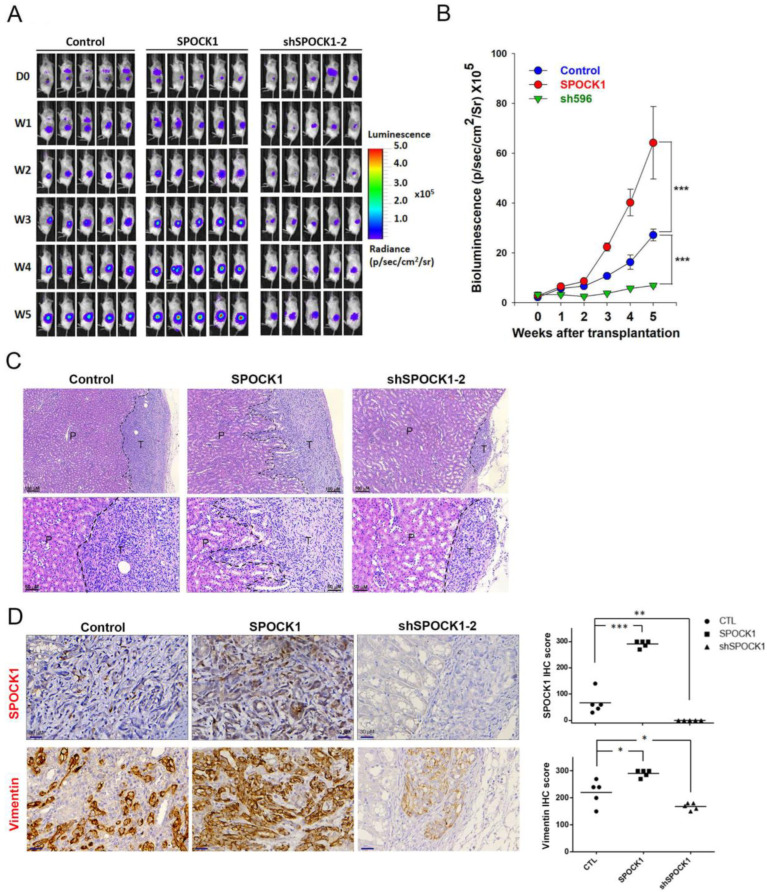
SPOCK1 regulates tumorigenicity and invasive ability in vivo. (**A**) Luciferase-tagged Cak-1 cells with stable SPOCK1-KD or overexpression were implanted into kidneys of NOD/SCID mice. Luciferase activity was detected at indicated time points with an IVIS imaging system. Representative bioluminescent images of mice from the control, SPOCK1-overexpression, and SPOCK1-KD groups taken per week are shown. (**B**) Quantitative analysis of Xenogen imaging signal intensity (photons/s/cm^2^/sr) every week. (**C**) The H&E staining of orthotopically implanted tumors shows the interface between the generated murine tumor and renal parenchyma. At the end of the study, tumor-injected kidneys were isolated and examined by the H&E staining. The interface between the generated murine tumor and renal parenchyma is indicated by the black dashed line, and parenchyma infiltration was observed in tumor cells overexpressing SPOCK1. Scale bar = 50 or 100 µm. (**D**) SPOCK1 and vimentin protein expression levels in Caki-1 orthotopic tumors expressing SPOCK1 or shSPOCK1 were measured by IHC staining. Scale bar = 30 µm. Right panel, quantification of the SPOCK1 and vimentin IHC staining score. * *p* < 0.05, ** *p* < 0.01, *** *p* < 0.001, compared to the control group.

**Figure 7 cells-12-00352-f007:**
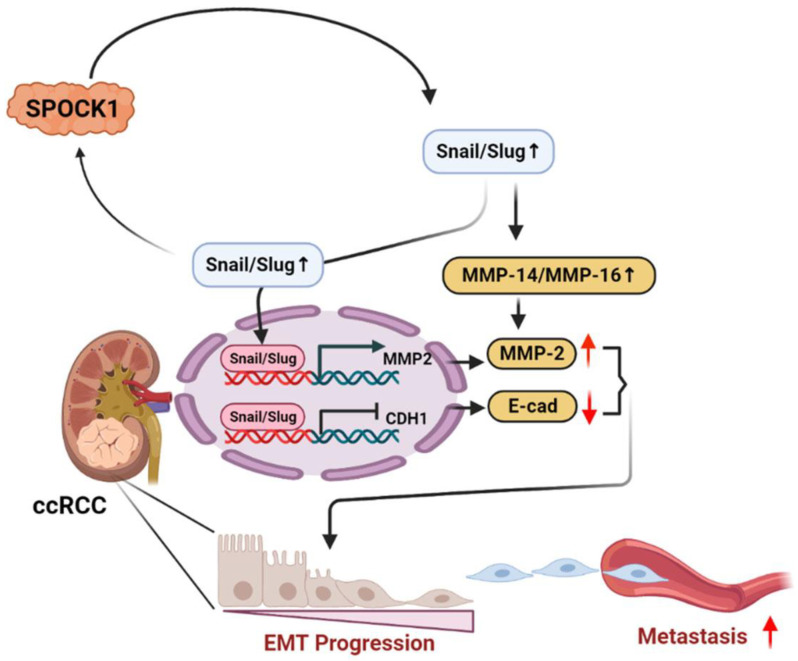
A working model shows the molecular mechanism underlying the ability of SPOCK1 to promote the progression of clear cell renal cell carcinoma (ccRCC) cells. The prometastatic effect of SPOCK1 on ccRCC cells was attributed to induce the upregulation of Snail family members (Snail and Slug) following transcriptional downregulation of E-cadherin and upregulation of the matrix metalloproteinase (MMP)-2. Moreover, SPOCK1-induced upregulation of Snail family members also shows the positive feedback regulation of SPOCK1. These signal pathways ultimately trigger epithelial-to-mesenchymal transition (EMT) progression and subsequent promotion of ccRCC metastasis. Green ovals indicate hypothetical regulators which might be involved in the SPOCK1-mediated interplay of Akt and Snail family members.

**Table 1 cells-12-00352-t001:** Cox univariate and multivariate regression analyses of prognostic factors and SPOCK1 expression for overall survival (OS) in 533 clear cell renal cell carcinoma (ccRCC) patients.

Variable	Comparison	HR (95% CI)	*p* Value
**Cox univariate analysis (OS)**			
Age	<61 years; ≥61 years	1.771 (1.305~2.405)	<0.001 *
Gender	Female; male	0.949 (0.698~1.292)	0.741
Stage	I/II; III/IV	3.833 (2.795~5.255)	<0.001 *
Tumorsize	T1/2; T3/4	3.151 (2.329~4.262)	<0.001 *
Nstatus	N0; N1	0.912 (0.785~1.059)	0.227
Mstatus	M0; M1	2.143 (1.706~2.692)	<0.001 *
SPOCK1	Low; high	1.630 (1.202~2.211)	0.002 *
**Cox multivariate analysis (OS)**			
Age	<61 years; ≥61 years	1.693 (1.238~2.315)	0.001 *
Stage	I/II; III/IV	3.626 (1.916~6.863)	<0.001 *
Tumorsize	T1/2; T3/4	0.785 (0.431~1.428)	0.427
Mstatus	M0; M1	1.809 (1.365~2.398)	<0.001 *
SPOCK1	Low; high	1.372 (1.003~1.876)	0.048 *

HR, hazard ratio; CI, confidence interval; N, node; M, metastasis. * *p* value of <0.05 was statistically significant.

**Table 2 cells-12-00352-t002:** Cox univariate and multivariate regression analyses of prognostic factors and SPOCK1 expression for disease-free survival (DFS) in 117 clear cell renal cell carcinoma (ccRCC) patients.

Variable	Comparison	HR (95% CI)	*p* Value
**Cox univariate analysis (DFS)**			
Age	<61 years; ≥61 years	1.222 (0.441~3.386)	0.7
Gender	Female; male	2.208 (0.702~6.940)	0.175
Stage	I/II; III/IV	3.440 (1.235~9.583)	0.018 *
Tumorsize	T1/2; T3/4	3.401 (1.222~9.471)	0.019 *
Nstatus	N0; N1	1.071 (0.634~1.808)	0.798
Mstatus	M0; M1	0.198 (0.003~15.515)	0.128
SPOCK1	Low; high	3.610 (1.146~11.367)	0.028 *
**Cox multivariate analysis (DFS)**			
Stage	I/II; III/IV	2.377 (0.815~6.934)	0.113
Tumorsize	T1/2; T3/4	NA	NA
SPOCK1	Low; high	2.922 (0.862~9.909)	0.085

HR, hazard ratio; CI, confidence interval; N, node; M, metastasis; NA, not available. * *p* value of <0.05 was statistically significant.

## Data Availability

All data generated or analyzed during this study are included in this article and its Appendix A.

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
