# Peer review of "Proteoglycan SPOCK1 as a Poor Prognostic Marker Promotes Malignant Progression of Clear Cell Renal Cell Carcinoma via Triggering the Snail/Slug-MMP-2 Axis-Mediated Epithelial-to-Mesenchymal Transition"

_cells, 2023, doi:10.3390/cells12030352_

Round 1
Reviewer 1 Report
For authors
The paper „Proteoglycan SPOCK1 as a poor prognostic marker promotes malignant progression of clear cell renal cell carcinoma via triggering the Snail/Slug-MMP-2 axis-mediated epithelial-to-mesenchymal transition” by Yung-Wei Lin and co workers is a study which unveils the role of SPOCK1 in metastatic progression in renal cell carcinoma by providing a molecular mechanism between SPOCK1 and EMT actors.
The study is well based from the experimental point of view. It uses appropriate methods and techniques for documenting the signaling pathways, key targets, cellular processes related to EMT in order to demonstrate the role of SPOCK1 in modulating this transition with implications in metastasis.
However, following its thorough analysis I consider the manuscript has to address minor and major revision.
1) The minor ones are related to expression of phrases, ideas from the English language point of view. In some case the statements, comments made in the text are not supported by data shown in the Figures. Please, make the suggested changes and provide the requested clarifications/ answers indicated at each point.
2) The major concerns are related to the following problems:
2a) The controls- The study relies extensively on experiments of gene downregulation and overexpression. In both cases the targeted proteins are compared with the control counterparts in sh-ctrls and Neo respectively.
It has been noticed that in RNAi experiments (which is also a system addresing gene downregulation), the so-called scrambles or controls, are not totally inert towards different genes (Anal Biochem. 606, 113828 (2020), Kleefeldt JM et al. Commercially available transfection reagents and negative control siRNA are not inert). In order to rule out such a possiblity, an absolute control, represented by cells w/o any treatment, but manipulated in paralel, exactely as the transfected samples would be required for both downregulation and overexpression experiments, for all analyzed targets.
For example the situation shown in Fig. 2, where there are visible differences between „control samples” . SPOCK1 expression in Caki-1/Neo (which is used as a control , Fig. 2E) is practically nule whereas in Caki-shCtrl (control, Fig.2C) is positive but visibly lower than in Caki-1/ccRCC – (the absolute control , Fig. 2A).
Until the impact of ”controls” used for downregulation or overexpression are shown comparatively with the untreated cells (the absolute control), all other results which are referred to them can not be accurately analyzed and addressed.
2b) Supplementary data- I did not find any comments related to Fig. 7S,8S, 9S in Results section. There are several assumptions based of these but in Discussion chapter. I would suggest these Figures which are experimental data to be embedded in the Results chapter, as are Fig1S-Fig.6S, where could be properly commented. If needed, they can be referred in the Discussion chapter as well.
Minor revisions
1) Introduction, lines 52-23 “Clear cell renal cell carcinoma (RCC; ccRCC) represents the most common (at >80%) form of RCC, and papillary (p)RCC is the second most common subtype of RCC (at 10%~15%).”
Since you use double abbreviation RCC or ccRCC immediately after Clear cell renal carcinoma it is not clear which one stands for this and will be used further. I assume you want to use RCC for renal cell carcinoma and ccRCC for Clear cell renal cell carcinoma.
In this case better say “ Renal cell Carcinoma (RCC) includes Clear cell renal cell carcinoma (ccRCC) which represents the most common (at >80%) form of RCC and papillary (p)RCC, the second most common subtype of RCC (at 10%~15%)”.
2) Introduction, line 75 Please revise the phrase, as is redundant. “As already reported, MMP-14 expression was reported” .
3) Introduction, line 82-83. Again, be careful how you address the content. All polypeptidic chains have an N-terminus and a C-terminus. It is of interest for the readers to mention this if you communicate some info about either N- or C-terminus or both. As you already did about N-terminus is ok. But simple stating that SPOCK1 has a C-terminus is of no interest. I would suggest to change this phrase.
“SPOCK1 and other members of the SPARC family share a similar 82 structure that consists of an N-terminus, follistatin-like domain, and C-terminus, and it plays crucial roles in cell proliferation, apoptosis, adhesion, and migration.”
with
„SPOCK1 shares similar structure with other members of the SPARC family, having an N-terminus with follistatin-like domain, and plays crucial roles in cell proliferation, apoptosis, adhesion, and migration”.
RESULTS
L 1) Line 260-262 –The authors state , on one hand that “There was no significant difference in SPOCK1 levels between the RCC group and normal group“ and on the other hand that ”significantly higher levels of SPOCK1 transcripts were observed in ccRCC tissues compared to normal tissues”. As ccRCC were part of the RCC group would not be rationale to expect that contribution of SPOCK1 from ccRCC to be reflected in the overall RCC when this is compared with normal group ? How do authors comment on this ?
2) Fig 1 D – Please explain in figure legend what represents the right-side panels. I asume there are enlaged images of the left-side ones.
3 3) In Fig. 2 are described the experiments related to expression, proliferation , clonogenicity, invasion and migration. In order to enssure that readers follow easily the data and commets it is better to describe an experiment all through before you move to the next one. And not coming back to the previous one. For ex. Fig 2A (lines 317-319), then you move to Fig.2B (lines 320-322), and you come back to Fig. 2A (Lines 323-324).
44)Lines 320-22 „Next, the migratory abilities of these cell lines were further checked, and we found a variety of migratory abilities, with Caki-1 and 786-O cells exhibiting relatively high migratory abilities (Figure 2B)”.
Again, please avoid repeating words in the same phrase; „migratory abilities –used three times in a row does not make the phrase clearar. What do you mean by „a variety of migratory abilities” ?
5) Line 326-327 „SPOCK1-KD was performed by three specific short hairpin (sh)RNAs to establish stable KD of endogenous SPOCK1 in Caki-1 and 786-O cells”.
Please explain what KD abreviation stands for ? If is KnockDown this should be introduced either in the 2.7 chapter or in the above text as SPOCK1-Knockdown (KD). And then keep the abreviation only (including in Figure legend).
6) Fig. 3D , lower panels. What is the difference between WBs representing snail, slug in shSPOCK1-2 (left panel) and snail and slug shown also in shSPOCK1-2 (right panel) ?
7) Fig. 3F Please explain GIPZ what stands for?
8) Lines 408,409 “We next screened the effects of SPOCK1 on these proteases and found that expressions of MMP-14, MMP-16, and its downstream targets, MMP-2 and MMP-9, were considerably downregulated after SPOCK1-KD in Caki-1 cells”.
Where are presented the results which support this statement?
9) Lines 412-413 “In contrast to Caki-1 cells, Western blot data showed that MMP-14, MMP-16, and MMP-2 were consistently suppressed by SPOCK1-KD in 786-O cells (Figure 4B)”. This statement is not supported entirely by data shown in Fig.4B. MMP14 and MMP16 are indeed lower in sh-SPOCK1-2 786-O cells than in sh-SPOCK1-2 Caki-1 cells. However , MMP-3 is visibly similar in both and MMP-2 is quite oposite, lower in sh-SPOCK1-2 Caki-1 than in sh-SPOCK1-2 786-O. Please, revise this text accordingly.
10) Fig. 4D In Figure 4 legend is mentioned “ (D) MMP-2 activity of SPOCK1-overexpressing Caki-1 and 786-O cells were detected by a gelatin zymographic assay”. However ,there are no comments in the text related to results presented in this. Just “In addition to protein expression, MMP-2 proteolytic activity was respectively upregulated and downregulated in SPOCK1-overexpressing and SPOCK1-depleted ccRCC cells compared to parental cells (Figure 4D, S6)“. Is it referred to Fig. 4D, S6 or to Fig.4D? The image for pro-MMP-2 or MMP2 in 786-O/SPOCK1 is practically nule compared with Neo, so the text is not is agreement with what zymography shows.
Please , either keep the statement only for Caki-1/SPOCK cells, or provide a convincing zymography for 786-O/SPOCK1 cells as well.
11) Lines 457 -458 “Herein, we observed that KD and overexpression of SPOCK1 in Caki-1 cells respectively inhibited and promoted MMP2 mRNA expression (Figure 5A)”. Please reformulate correctly “Herein, we observed that KD and overexpression of SPOCK1 in Caki-1 cells inhibited and promoted MMP2 mRNA expression respectively (Figure 5A)”.
12) Lines 467-471 “Moreover, SPOCK1-KD-mediated downregulation of N-cadherin and vimentin and upregulation of E-cadherin were dramatically reversed by overexpressing MMP-2 (Figure 5F), indicating that Snail family-mediated MMP-2 expression participates in SPOCK1-induced EMT progression”.
Please explain what represent the lane shSPOCK1-2/MMP2 and lane MMP2 in Fig 5F? What is the difference between them.
Discussion
13) Lines 533-“ OS, DSS, and DFS times” stands what for?
14) Lines 561-562 “Our present study also demonstrated that recombinant SPOCK1 treatment could induce Akt activation in Caki-1 and 786-O cells (Figure S7)”. This is related to experimental results and has to be presented in Results section.
The same for Figures S8, S9.
Reviewer 2 Report
In general, the present work is good and can be considered to publish in the target journal. Two comments are as following, one is that it is better to remove Akt from the schematic graph, since this is not investigated. The other one is that statistics can be provided in Fig. 6C & D.
Reviewer 3 Report
The manuscript by Lin et al entitled “Proteoglycan SPOCK1 as a poor prognostic marker promotes malignant progression of clear cell renal cell carcinoma via triggering the Snail/Slug-MMP-2 axis-mediated epithelial-to-mesenchymal transition” examines the role of SPOCK1 in the progression of clear cell renal cell carcinoma (ccRCC). The authors claim that SPOCK1 alone can be used as ccRCC predictive marker in patients. The topic is of great interest to the community and would promote significant support for future research in cancer. However, the manuscript title, graphical abstract, and conclusion overstate the impact of their actual findings. The authors use insilico analysis, gene enrichment, and pathways to depict the main conclusions which are based on the whole sample, not individual cellular components. At this point, no definite cause-effect relationships have been done or shown in the current manuscript to claim such findings. Results from in vitro, however, do show that SPOCK1 may affect the progression in ccRCC cell lines. But this result alone does not imply that SPOCK1 or expression-correlated genes affect EMT progression and ccRCC metastasis. It would be of a great interest if the authors could replicate the tests in vivo as well. This would significantly improve the manuscript and would provide evidence that SPOCK1 may play a role in ccRCC progression.
Other issues
1. I would like to observe the correlation values they obtained from SPOCK1 in TCGA-KIRK data.
2. While the authors have conducted GSEA using TCGA-KIRK datasets, there was not a clear statement of choosing such data among other RCC datasets in cBIOportal database. I would like to hear from authors about such preferences.
3. Graphical abstract is misleading. The research in this manuscript does not provide proof that SPOCK1 has any direct relation with metastasis.
4. The manuscript needs a careful English-proof edition.
Round 2
Reviewer 3 Report
The authors could address my questions and issues properly. I do not have any other comments to add or suggestions to make.